# Assessing SATNet's Ability to Solve the Symbol Grounding Problem

**Oscar Chang, Lampros Flokas, Hod Lipson**
Data Science Institute
Columbia University
{oscar.chang,lampros.flokas,hod.lipson}@columbia.edu

**Michael Spranger**
Sony AI
michael.spranger@sony.com

## Abstract

SATNet is an award-winning MAXSAT solver that can be used to infer logical rules and integrated as a differentiable layer in a deep neural network [1]. It had been shown to solve Sudoku puzzles visually from examples of puzzle digit images, and was heralded as an impressive achievement towards the longstanding AI goal of combining pattern recognition with logical reasoning. In this paper, we clarify SATNet's capabilities by showing that in the absence of intermediate labels that identify individual Sudoku digit images with their logical representations, SATNet completely fails at visual Sudoku (0% test accuracy). More generally, the failure can be pinpointed to its inability to learn to assign symbols to perceptual phenomena, also known as the symbol grounding problem [2], which has long been thought to be a prerequisite for intelligent agents to perform real-world logical reasoning. We propose an MNIST based test as an easy instance of the symbol grounding problem that can serve as a sanity check for differentiable symbolic solvers in general. Naive applications of SATNet on this test lead to performance worse than that of models without logical reasoning capabilities. We report on the causes of SATNet's failure and how to prevent them.

## 1   Introduction

Machine learning systems have become increasingly capable at a wide range of tasks, with neural network based models outperforming humans at tasks like object recognition [3], speech recognition [4, 5], the game of Go [6, 7], Atari videogames [8, 9], and more. Nonetheless, the success of deep learning comes with significant caveats: neural networks require immense amounts of labeled data for training, can be easily tricked by tiny input perturbations or spurious correlations, and succumb to brittle generalization when tested on data that deviate ever so modestly from the training distribution. Critics point to these caveats as evidence that deep learning, in its current incarnation, is really just performing a sophisticated type of pattern matching, the likes of which can only ever constitute intelligence in narrow, circumscribed domains [10, 11].

By comparison, human intelligence can be applied more generally. This has been argued to be a result of two distinct modes of cognition: *System 1* and *System 2* [12, 13]. System 1 happens quickly and without conscious effort, for example comparing the size of objects or locating the general source of a sound. On the other hand, System 2 involves slow and deliberate attention, for example solving for a complicated arithmetic equation or checking that an argument is logical. Current machine learning systems have been likened to System 1 [14], because System 1 mostly involves the use of associative

memory, and is highly susceptible to cognitive biases and sensory illusions. Symbolic AI algorithms that are based on logic and search more closely resemble System 2.

To achieve robust human-level AI that can solve non-trivial cognitive tasks, it is crucial to combine both System 1 like *pattern recognition* and System 2 like *logical reasoning* capabilities in a seamless *end-to-end learning* fashion. This is because in many practical problems of interest, it is difficult and expensive to collect intermediate labels to train specific machine learning sub-components. For example, it appears infeasible to build a 'danger' classifier for a self-driving car, where every possible dangerous scenario is pre-determined and categorized beforehand. Researchers are thus far able to combine both capabilities in a single AI system, but not train them end-to-end. Famously, OpenAI's very impressive achievement of controlling a robotic hand to solve a Rubik's cube required the separate use of a machine learning system to perform the dexterous manipulation and a discrete solver to decide the side of the cube that should be turned [15].

Attempts to bridge the two capabilities seamlessly belong to one of three approaches. The first involves augmenting deep learning models with soft logic operators [16–22] or combinatorial solving modules [23–27]. However, this approach typically requires the programmer to pre-specify intricate logical structures according to the problem domain. Moreover, these logical components are fixed and not amenable to learning. The second approach uses sub-symbolic reasoning techniques like Recurrent Relation Networks to implicitly pick up on logical structures within the problem [28–30]. This approach improves on the first by learning the logical structure implicitly by optimization, but nevertheless also necessitates careful feature engineering. The third approach is the field of inductive logic programming (ILP), which starts from a traditional symbolic AI model like a knowledge base, and adds learning capabilities to it [31–34]. Unfortunately, ILP is limited to symbolic inputs and outputs, unlike deep neural networks.

Against the backdrop of such approaches, SATNet [1] promised to integrate "logical structures within deep learning" with a differentiable MAXSAT solver that can infer logical rules and be used as a neural network layer. SATNet claimed to have solved problems that were "impossible for traditional deep learning methods and existing logical learning methods to reliably learn without any prior knowledge," most notably solving a Sudoku puzzle visually from images of puzzle digits, and was awarded with a Best Paper Honorable Mention at 2019's *International Conference on Machine Learning*.

Based on SATNet's success, one might think that enabling end-to-end gradient-based optimization (i.e. making every component in a system differentiable) is sufficient for end-to-end learning (i.e. learning without intermediate supervision signals). However, defining gradients for an objective does not, on its own, result in successful learning outcomes, as exemplified by the history of deep learning. Successful training of architectures with hundreds of layers, where gradients are trivially well defined, is highly non-trivial and requires careful initialization, batch normalization, adaptive learning rates, etc. Additionally, without an appropriate inductive bias (like the rules of the game), learning to solve complex problems like visual Sudoku from relatively few samples is extraordinarily challenging. It is unlikely that end-to-end gradient-based optimization by itself will, in general, result in models that generalize well.

Thus, SATNet's claim to have solved the end-to-end learning problem of visual Sudoku "in a minimally supervised fashion" should be revisited. **Can SATNet learn to assign logical variables (symbols) to images of digits (perceptual phenomena) without explicit supervision of this mapping?** This is also known as the symbol grounding problem [2], which has long been thought to be a prerequisite for intelligent agents to perform real-world logical reasoning. If answered in the affirmative, SATNet would have marked a revolutionary leap forward for the whole field of AI, by virtue of the difficulty of the symbol grounding problem in visual Sudoku.

The general complexity of the symbol grounding problem embedded in end-to-end learning should not be underestimated. Figure 1 directly exemplifies the difficulty of the symbol grounding problem for both human and artificial intelligence. Common measures of abstract reasoning in artificial intelligence such as DeepMind's PGM work similarly to Raven's Progressive Matrices (a test for human intelligence), where predicting what comes next involves determining the hidden attributes (symbols) in what has been presented (perceptual phenomena), and inferring the pattern from them [11, 35–37]. Once given the hidden attributes, it is trivial for a human or a combinatorial solver to infer the pattern [35]. However, jointly inferring the hidden attributes together with the pattern proves to be a challenging cognitive task in general.

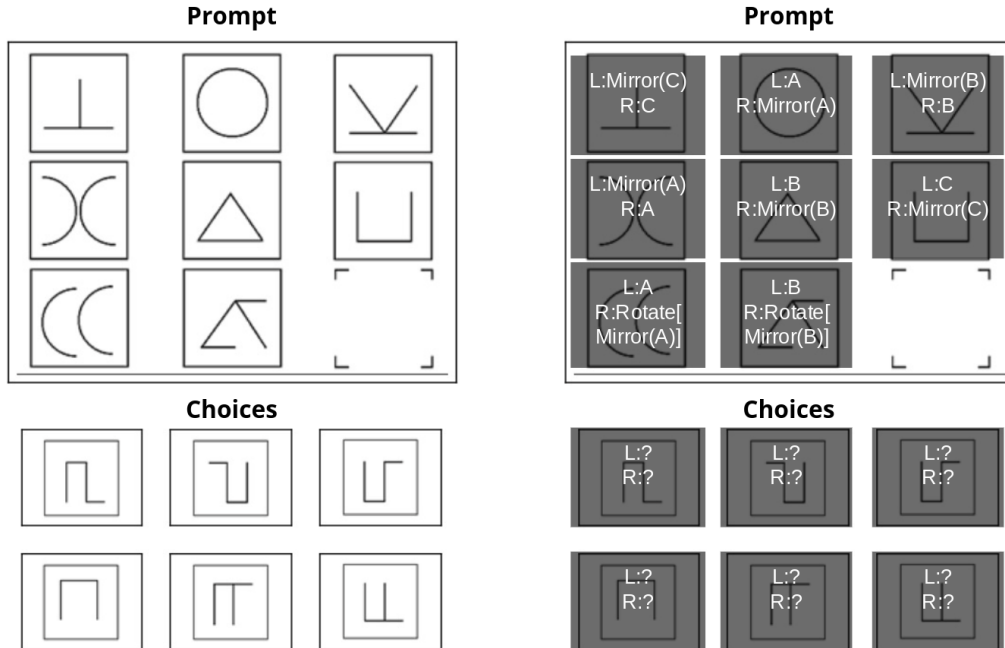

Figure 1: A challenging Raven's Matrix puzzle that exemplifies a difficult instance of the symbol grounding problem. We invite the reader to attempt the puzzle for themselves on the left hand side of the figure first, before looking at the annotations on the right hand side. Once the given images have been decoded to an appropriate symbolic representation, it is straightforward for a discrete solver or a human to solve it. For a full explanation of the solution, please see Appendix Section A.

## 1.1 Our Contribution

In this paper, our principal contribution is a re-assessment of SATNet that clarifies the extent of its capabilities and a discussion of practical solutions that will help future researchers train SATNet layers in deep networks.

First, we observed from the SATNet authors' open-source code that intermediate labels are leaked in the SATNet training process for visual Sudoku. The leaked labels essentially result in a two-step training process for SATNet, where it first uses the leaked labels to train a digit classifier, and then uses the symbolic representations of the digits to solve for the Sudoku puzzle. After removing the intermediate labels, SATNet was observed to completely fail at visual Sudoku (0% test accuracy). If intermediate labels are available, it is possible to separately pre-train a digit classifier and then use SATNet, independent of a deep network, to solve for the puzzle. This might even be preferable, given our finding that SATNet fails in 8 out of 10 random seeds despite access to the labels, which is evidence that SATNet struggles to learn to ground the Sudoku digits into their symbolic representation. To be clear, the label leakage did not affect SATNet in the non-visual case, and its success on purely symbolic inputs and outputs nonetheless marks progress in ILP, but does not fix the field's persisting deficiency in dealing with perceptual input.

While solving *difficult* instances of the symbol grounding problem like visual Sudoku or PGM might be beyond the reach of SATNet, we found that SATNet also cannot solve *easy* instances, unless properly configured. We devised a test called the *MNIST mapping problem*, whose solution requires merely digit classification (a simple problem for neural networks) and learning a bijective mapping between logical variables (a simple problem for discrete solvers). This test serves as an easy instance of the symbol grounding problem, and is suitable as a sanity test not just for SATNet, but other prospective differentiable symbolic solvers. Even on a simple test like this, a naive application of SATNet can cause it to perform worse than models without logical reasoning capabilities.

Our work identifies several factors that affect the learning dynamics of SATNet and provides practical suggestions for configuring SATNet to enable successful training. We reveal surprising complexities that are unique to SATNet and break standard deep learning norms. For example, using different

learning rates for different layers in neural networks is not a common practice, since the use of Adam usually suffices. But for the case of SATNet, even when Adam is used, the backbone layer has to learn at a slower rate than the SATNet layer for successful training to occur. Surprisingly, we found that unconditionally increasing the number of auxiliary variables does not increase the expressivity of the model, but instead leads to a complete failure in learning. Further adjusting the choice of optimizer and neural architecture led to statistically significant improvements, culminating in near perfect test accuracy (99%).

The rest of the paper is organized as follows: Section 2 reviews the relevant technical background for SATNet and visual Sudoku. Section 3 examines the subtle nature of the label leakage in the original SATNet paper and its ramifications. Section 4 describes the MNIST mapping problem, and investigates optimal SATNet configurations for this simple MNIST-based test. Finally, we conclude in Section 5.

## 2 Background

### 2.1 SATNet

SATNet is a neural network layer that solves a semidefinite programming (SDP) relaxation of the following MAXSAT problem,

$$\max_{\tilde{v} \in \{-1,1\}^n} \sum_{j=1}^{m} \bigvee_{i=1}^{n} \mathbf{1} \left\{ \tilde{s}_{ij} \tilde{v}_i > 0 \right\}, \tag{1}$$

where $\tilde{v} \in \{-1,1\}^n$ denotes assignments to $n$ binary variables, and $\tilde{s}_i \in \{-1,0,1\}^m$ denotes the sign of variable $\tilde{v}_i$ in $m$ clauses. The set of $\tilde{s}_{ij}$, denoted by $S$, forms the SATNet layer's learnable parameters. $\tilde{v}$ can be partitioned into two disjoint sets $\mathcal{I}$ and $\mathcal{O}$, which are represented in SATNet by layer inputs $Z_{\mathcal{I}}$ and outputs $Z_{\mathcal{O}}$ (which can be either probabilistic or strictly binary), and their respective continuous relaxations $V_{\mathcal{I}}$ and $V_{\mathcal{O}}$. Gradients from the layer output $\nabla_{Z_{\mathcal{O}}} \mathcal{L}$ are backpropagated to both the layer's weights in the form of $\nabla_S \mathcal{L}$ and to the layer input in the form of $\nabla_{Z_{\mathcal{I}}} \mathcal{L}$. The two main tunable hyperparameters in a SATNet layer are the number of clauses $m$ and the number of auxiliary variables $aux$ (which "play a role akin to register memory that is useful for inference"). Auxiliary variables are also input variables, but unlike $Z_{\mathcal{I}}$, they are not the output of preceding layers.

### 2.2 Visual Sudoku

Sudoku is a number puzzle played out on a 9-by-9 grid. Each of the 9x9=81 cells has to contain a digit from 1 to 9. The game starts out from a partially filled grid, and the object of the game is to complete the rest of the cells on the grid. Each of the digits from 1 to 9 has to appear exactly once in every row, column, and each of the nine 3-by-3 subgrids. In the *non-visual* case, the state of the Sudoku grid can be encoded using 9x81=729 binary variables, and SATNet can learn to map from the binary encoding of the initial grid to the binary encoding of the completed grid without the programmer having to explicitly encode for the rules of the game. Given 9000 training and 1000 test examples (with 36.2 pre-filled cells on average), where each example is a pair consisting of the initial and completed grid, SATNet achieves 99.7% training and 98.3% test accuracy. By comparison, a symbolic solver that knows the rules of the game can provably solve the game perfectly [38], while a purely deep learning based approach, trained on a million examples, scores 70.0% on a test set of thirty games [39]. We report on other related work on non-visual Sudoku in Appendix Section B.

In *visual* Sudoku, the inputs are now 81 images of digits (taken from the MNIST dataset), with '0' standing in for empty cells. They are processed by a convolutional neural network (CNN) backbone with a SATNet layer, which performs at 93.6% training and 63.2% test accuracy using the same number of training and test examples. The SATNet authors contextualized their findings by claiming that the "theoretical best" test accuracy is capped at 74.8% ($\approx 0.992^{36.2}$), which is the probability that the LeNet[1] CNN backbone, which has 99.2% test accuracy on MNIST, has correctly classified all the pre-filled cells.

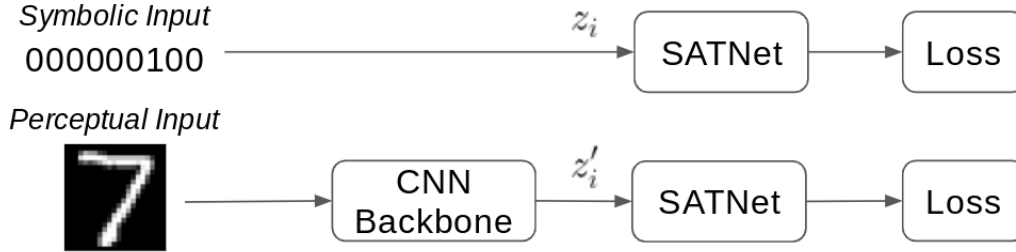

Figure 2: A visualization of the difference between symbolic and perceptual inputs.

# 3 SATNet Fails at Symbol Grounding

## 3.1 The Absence of Output Masking

While every Sudoku puzzle corresponds to 729 logical variables in the MAXSAT problem (excluding the auxiliary variables for now), the number of pre-filled cells and their positions differ depending on the puzzle. Thus, $\mathcal{I}$ and $\mathcal{O}$ are different for each example, even though the sizes of $Z_{\mathcal{I}}$ and $Z_{\mathcal{O}}$ are fixed beforehand and not example-dependent. A straightforward way to solve this is to apply an appropriate bit mask depending on the example.

Consider a toy example with 5 variables $v_1 = 1, v_2 = 0, v_3 = 0, v_4 = 1, v_5 = 0$ where $\mathcal{I} = \{1, 2, 3\}$ and $\mathcal{O} = \{4, 5\}$. Then, the input to SATNet should be 10000 with the bit mask 11100, and the output should be 00010 with the bit mask 00011. The problem with the original SATNet implementation is that the bits that correspond to the inputs are not masked in the output.

Not masking the output might not seem problematic, given that SATNet does not modify input variables $Z_{\mathcal{I}}$ nor their relaxations $V_{\mathcal{I}}$. But consider the decomposition of the loss function $\mathcal{L}$ into a sum of binary cross entropies (BCE) between the SATNet variables $z$ and the training label $l$.

$$\mathcal{L} = \sum_{i=1}^{n} \text{BCE}(z_i, l_i) = \sum_{i \in \mathcal{I}} \text{BCE}(z_i, l_i) + \sum_{o \in \mathcal{O}} \text{BCE}(z_o, l_o). \tag{2}$$

Since the $z_{\mathcal{I}}$ are not modified by SATNet, $z_i = l_i$ for $i \in \mathcal{I}$, effectively zero-ing out any loss contributed by terms in $z_{\mathcal{I}}$. This is true when SATNet is applied to purely symbolic problems like non-visual Sudoku.

However, once perceptual input is introduced, $z_i$ is not directly accessible by SATNet. Instead, the input to the SATNet layer is a symbolic representation $z_i'$ of features extracted from the data (see Figure 2). Thus, the loss from $z_{\mathcal{I}}$ in Equation 2 is non-zero before the neural network has learned to ground the symbols appropriately, i.e. $z_i' = z_i = l_i$. Not masking the output to SATNet thus leaks label information to the layers before the SATNet layer, effectively training a classifier that learns to map from the perceptual data to the appropriate symbol representation, i.e. symbol grounding.

## 3.2 Visual Sudoku

Table 1: Effects of Output Masking

| Accuracy | Non-Visual Sudoku | | Visual Sudoku | |
|---|---|---|---|---|
| | Original | Masked Outputs | Original | Masked Outputs |
| Train | 99.7±0.0% | 99.7±0.0% | 18.5±12.3% | 0.0±0.0% |
| Test | 97.6±0.1% | 97.6±0.1% | 11.9±7.9% | 0.0±0.0% |

We re-ran the Sudoku experiments using the SATNet authors' open-sourced implementation with identical experimental settings, but over 10 different random seeds to get standard error confidence intervals. Table 1 shows clearly that output masking does not affect the results in the non-visual case, but causes SATNet to fail completely for visual Sudoku, which is what we expect from the discussion in the previous section. Once the intermediate labels are gone, the CNN does not ever

learn to classify the digits better than chance. SATNet's failure at symbol grounding directly leads to its failure at the overall visual Sudoku task.

Interestingly, we also found that SATNet's performance in visual Sudoku in the absence of output masking is highly dependent on the random initialization, with 8/10 random seeds leading to complete failure as well. This explains why SATNet's performance over 10 runs (18.5% training accuracy) is dramatically lower than what was originally reported (93.6% training accuracy). Therefore, even for problems where we have access to intermediate labels, leaking them indirectly via the absence of output masking is strictly less desirable than directly pre-training a neural network classifier with those labels. In Section 4.1, we note important strategies for mitigating complete failure.

Of the 2 runs that succeeded (i.e. had non-zero training accuracy, specifically 93.2% and 91.7% respectively), we found that the label leakage basically results in a two-step training process for SATNet, where the CNN first learns to do MNIST digit classification, and then the SATNet layer learns to solve the actual Sudoku problem. We show in Figure 3 training accuracy plots of two example runs, one successful and the other not. They are annotated with corresponding plots (at the bottom for comparison) of the CNN's classification accuracy on the MNIST test set. For the successful runs, we observe that the training accuracy for visual Sudoku stays at zero for a small number of epochs, during which time the leaked labels help train the CNN to be an MNIST digit classifier. Only after the digit classifier works to some degree, does the training accuracy for visual Sudoku actually become non-zero. By contrast, in most of the unsuccessful runs, the CNN takes a very long time to become somewhat proficient at digit classification, and even after it does so, the SATNet layer seems unable to adapt to it, resulting in a permanent plateau at 0% training accuracy.

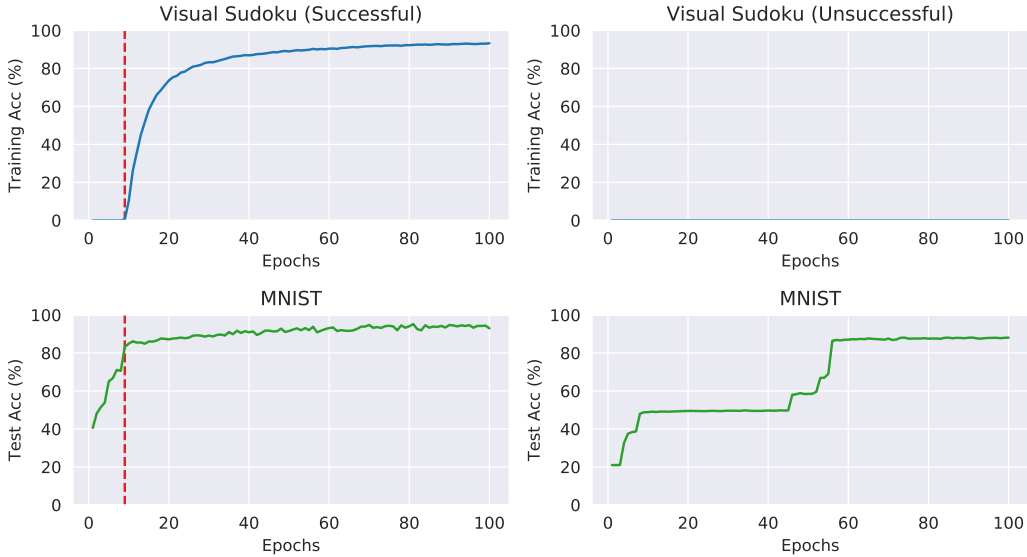

Figure 3: The graphs on the left show a successful run of SATNet on visual Sudoku, while the graphs on the right show an unsuccessful run. The successful run in the absence of output masking leads to a two-step training process, where the CNN first rapidly learns to classify digits, and then the SATNet layer learns to solve for Sudoku. The red vertical dotted line demarcates the point at which the training accuracy for visual Sudoku becomes non-zero. Unsuccessful runs typically take a long time for the CNN to classify digits, and never does better than 0% training accuracy at the overall visual Sudoku task.

## 4 MNIST Mapping Problem

The MNIST mapping problem involves a symbolic problem with 20 variables $v_i$, where the first ten variables are input (i.e. $\mathcal{I} = \{1, \ldots, 10\}$), and the next ten are output (i.e. $\mathcal{O} = \{11, \ldots, 20\}$). But the $v_{\mathcal{I}}$ are not provided directly; instead the input is given as perceptual data in the form of an MNIST digit image, and the challenge is to map an image of digit $i$ to the variable $v_{11+i}$. We assume that these variables are boolean (or the probabilistic equivalent, i.e. random variables taking real values in $[0, 1]$), but this should be adapted accordingly to the symbolic representation of a given solver.

There are two distinct sub-problems. The first sub-problem involves classifying an MNIST digit image into $v_1, \ldots, v_{10}$ (using a neural network). The second sub-problem involves learning a bijection (or an equivalent permutation) to $v_{11}, \ldots, v_{20}$ (using a symbolic solver), from which the class of the input image has to be identified. Both sub-problems taken on their own are considered to be *easy* problems. MNIST digits can be easily classified to 99% test accuracy [40], while permutation groups under equivalence queries are known to be exactly learnable in polynomial time [41]. Hence, we propose that a suitable sanity test for a differentiable symbolic solver is to solve the MNIST mapping problem to an accuracy of 99%. Note that a model that does not have to learn the bijection can circumvent the symbol grounding problem entirely by simply learning the output labels directly. Therefore, the test is strictly intended to be a check for symbol grounding, rather than a grand AI challenge that necessitates the combination of pattern recognition and logical reasoning as in visual Sudoku or PGM.

## 4.1 Configuring SATNet Properly

Surprisingly, some SATNet configurations fail the test, not by a slight margin, but completely (i.e. test accuracy no better than chance; we count them using 12% as a threshold to account for variance). In general, we found that the successful training of SATNet can be very sensitive to specific combinations of hyperparameters, optimizers, and neural architectures. We present four empirical findings using experiments on the MNIST mapping problem. All experiments were ran for 50 training epochs over 10 random seeds to get standard error confidence intervals. The Sudoku CNN, which was the backbone architecture used in the SATNet author's visual Sudoku implementation, is used throughout unless stated otherwise. We evaluate the results by presenting test accuracies with their confidence intervals and the number of complete failures in parentheses. For comparison, a non-SATNet baseline, which consists of the Sudoku CNN but with the SATNet layer replaced by two fully connected layers (1000 hidden units and ReLU), performs at 72.1±13.3% (3). At a minimum, SATNet should perform better than that, since its raison d'être disappears if it can be bested by equivalent models without logical reasoning capabilities.

**Finding 1** *Too little "logic" (i.e. low $m$) or too much "slack" (i.e. high $aux$) can cause failure.*

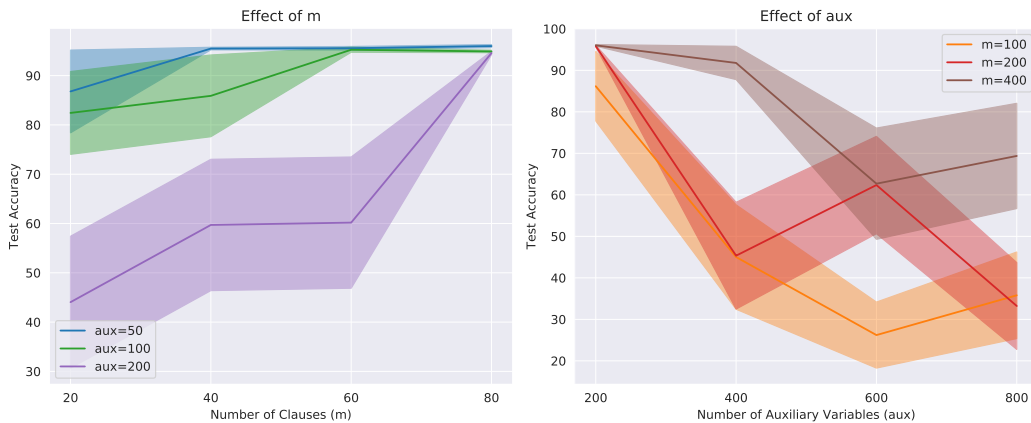

Figure 4: Both graphs show test accuracy on the MNIST mapping problem with the shaded interval representing the standard error.

The number of clauses $m$ controls the capacity of SATNet (rank of clause matrix), and we found that it can cause failure or result in terrible test accuracy when it is too low relative to what is needed for the problem. The number of auxiliary variables $aux$ also controls model capacity, but we observed that if it is too high for a given $m$, it can also cause failure (because most of the clauses end up being filled with meaningless input-independent auxiliary variables). High $m$ or low $aux$ do not affect test accuracy on the MNIST mapping problem, but they affect the amount of compute the SATNet layer uses.

**Finding 2**   *The backbone layer has to learn at a slower rate than the SATNet layer.*

Table 2: Effects of Different Learning Rates on the SATNet and Backbone Layer on Test Accuracy

| SATNet Layer | Backbone Layer Learning Rate | | |
|---|---|---|---|
| Learning Rate | $1\times10^{-3}$ | $1\times10^{-4}$ | $1\times10^{-5}$ |
| $1\times10^{-3}$ | $19.9\pm8.6\%$ (9) | $90.0\pm8.7\%$ (1) | $96.3\pm0.2\%$ (0) |
| $1\times10^{-4}$ | $17.4\pm4.3\%$ (8) | $74.6\pm8.6\%$ (0) | $96.1\pm0.2\%$ (0) |
| $1\times10^{-5}$ | $14.8\pm3.6\%$ (9) | $31.7\pm7.1\%$ (5) | $72.4\pm5.3\%$ (0) |

Table 2 shows the effect of differential learning rates between the SATNet and CNN backbone layers on test accuracy and number of failures, using Adam [42] for both layers. If the backbone layer has a higher learning rate than the SATNet layer, this often leads to failure. Optimal performance is observed when the backbone layer has a lower learning rate than the SATNet layer. Note that this might be counter-intuitive, given that in the label leakage scenario, the backbone CNN had to learn digit recognition before the SATNet layer could learn to solve Sudoku. But without label leakage, having a higher learning rate for the backbone does not make sense because it cannot learn anything useful without the help of the SATNet layer.

**Finding 3**   *Optimizing the backbone layer with SGD and the SATNet layer with Adam improves both training and test accuracy.*

Instead of simply using different learning rates, swapping the optimizer for the backbone layer with SGD raises test accuracy from $96.3 \pm0.2\%$ (0) to $98.6\pm0.1\%$ (0) (similarly so for training accuracy).

**Finding 4**   *A sigmoid output layer for the backbone is preferable to softmax.*

Table 3: Effects of Different Neural Architectures on Test Accuracy

| Architectures | Parameters | Backbone Output Layer | |
|---|---|---|---|
| | | Softmax | Sigmoid |
| LeNet [40] | 68,626 | $63.3\pm14.1\%$ (4) | $98.8\pm0.0\%$ (0) |
| Sudoku CNN | 860,780 | $98.6\pm0.1\%$ (0) | **$99.1\pm0.0\%$ (0)** |
| ResNet18 [43] | 11,723,722 | $67.6\pm6.3\%$ (0) | $97.2\pm0.9\%$ (0) |

The output of the CNN backbone has to take real values in $[0, 1]$; the SATNet authors' implementation used a softmax output layer to achieve this. We found that a sigmoid output layer strictly outperforms a softmax layer in all three architectures tested. When softmax is used, we observed that the size of the architecture can result in poor performance if it is too small or too big. In the case where it is too big, it is possible for accuracy to degrade rapidly after reaching its peak (we don't use early stopping). Of the three, the Sudoku CNN appears to be the optimal size.

Every model we tested failed at visual Sudoku, demonstrating the non-triviality of visual Sudoku's grounding problem (since getting even one puzzle in the test set correct necessitates the accurate classification of 36.2 digits on average). However, even for a seemingly easy instance of the symbol grounding problem in the form of MNIST mapping, it was highly non-trivial to find the correct SATNet configuration that would lead to 99% test accuracy. This shows that the current state of SATNet falls significantly short of its promise to integrate logical reasoning in deep learning.

## 5   Conclusion

In this paper, we presented a detailed analysis of SATNet's capabilities, and provided practical solutions that will help future researchers train SATNet layers in their deep neural networks more effectively. Specifically, we noted that the original experimental setup for visual Sudoku resulted in intermediate label leakage. After removing the intermediate labels, SATNet is found to completely fail at the task of visual Sudoku due to its inability to ground the images of the puzzle digits into

the appropriate symbolic representation. We further introduced the MNIST mapping problem as an easier instance of the symbol grounding problem compared to visual Sudoku, and found that SATNet needs to be delicately configured for training to be successful. In particular, the number of auxiliary variables cannot be increased unconditionally with respect to the number of clauses, and the backbone layer has to learn at a slower rate than the SATNet layer.

We can apply what we have learned about SATNet and its failure to solve visual Sudoku's symbol grounding problem more generally to other attempts to integrate logical reasoning into deep learning. Given that logical reasoning modules act at a symbolic level, while generic deep learning modules act at a sub-symbolic level, the interface between these two levels has to involve a symbol grounding problem. Hence, even if the intermediate label leakage identified in this paper might be SATNet-specific, we think that explicit tests against simple, interpretable instances of the symbol grounding problem will be fruitful for future researchers in discerning their claims about end-to-end learning (versus end-to-end gradient-based optimization).

In general, we think that the differences between deep learning and logic mirror the ones between continuous and discrete optimization. These differences go far deeper than the superficial lack of derivatives in discrete optimization, and we believe true progress has to come from significantly tighter integrations between deep learning and logic. We are excited that our work brings these differences to the forefront and encourages the community to think more critically about how to go about integrating logical reasoning into deep learning.

## Broader Impact

**Reproducibility**   In recent years, there has been a reproducibility "crisis" in the natural sciences and medicine [44–46], with the problem even extending into the computational sciences like machine learning [47–50]. There is little incentive for independent researchers to put in the effort to re-verify the claims of a paper that has gone through peer review. This is not least because of the possibility that the failure in replication might be due to problems with the replication rather than problems with the original claims. However, we believe that prominent papers, especially ones like SATNet that have won conference awards, deserve extra scrutiny. By re-assessing SATNet's original claims, we provide additional credibility for established findings in the machine learning literature. Sober re-assessments of cutting edge AI technology also help to downplay the 'hype', allowing non-expert stakeholders from the broader society to be clear-eyed about the current state of the art. We regret if this paper appears overly critical of the impressive achievements made by SATNet. A potentially negative consequence of our paper is that it discourages researchers from making their code open-source because of the additional scrutiny that this will invite. Critical assessments of AI technology might also lower both public and commercial funding for AI due to more realistic expectations, as has happened during the AI winters.

**The Importance of the Symbol Grounding Problem**   There have been many attempts to combine pattern recognition and logical reasoning into a single neural network model, but most of these attempts essentially focus on reducing the problem to the relaxation of non-differentiable functions. Our work on SATNet clearly exemplifies that addressing the optimization issues inherent in combining logic and deep learning will not be enough to train models in a minimally supervised end-to-end learning fashion. Without a significant breakthrough, solving symbol grounding problems without intermediate labels will probably remain out of reach. Our work aims to highlight the importance of explicitly addressing the symbol grounding problem, and we hope that future research to do so will expand the applications of machine learning and AI beyond System-1 pattern recognition capabilities.

## Funding Disclosure

The work in this paper was done when the first author was an intern at Sony AI.

## Footnotes

[1]To be precise, the SATNet authors used a bigger version with ∼10x more parameters than the original.

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
