[Supplementary Material]

# Appendix

## A  Solution to the Raven's Matrix puzzle

Figure 5: The three basic glyphs are formed from half a circle, a triangle, and a rectangle respectively.

Figure 6: The solution to the Raven's Matrix puzzle is the choice on the top right.

The source of this puzzle and its solution is user265554 [51] on Puzzling Stack Exchange.

Each panel is composed of a glyph on the left hand side (L) and a glyph on the right hand side (R). There are three basic glyphs (see Figure 5): a crescent (A), a half triangle (B), and a half rectangle (C). Each glyph can also be mirrored (Mirror), i.e. flipped horizontally, or rotated by 180 degrees (Rotate). In Figure 6, we annotate every panel in both the prompt and the choices with the symbols that represent it. It is clear that the blank in the prompt should be filled by a left glyph C and a right glyph Rotate[Mirror(C)], which is the choice on the top right.

## B  Related Work on Non-Visual Sudoku

On a dataset with 216,000 puzzles split in a 10:1:1 train-val-test ratio, a deep (recurrent relational) network that has access to positional information for each cell scores 100% test accuracy on puzzles with 33 pre-filled cells and 96.6% on puzzles with 17 pre-filled cells [29]. Amos and Kolter [27] use a differentiable quadratic programming layer called OptNet, which like SATNet has no a priori knowledge of the rules, in a neural network to solve for Sudoku. OptNet does not scale well computationally and can only solve 4-by-4 Sudokus.

# C  Experimental Settings

In the Supplementary materials, we provide source code and the shell commands to replicate all the experimental results in the paper.

## C.1  SATNet Fails at Symbol Grounding

The experimental settings for SATNet in Section 3 are identical to the original paper and based on the authors' open-sourced implementation available at `https://github.com/locuslab/SATNet`. Specifically, the CNN used is the sequence of layers: *Conv1-ReLU-MaxPool-Conv2-ReLU-MaxPool-FC1-ReLU-FC2-Softmax*, where *Conv1* has a 5x5 kernel (stride 1) and 20 output channels, *Conv2* has a 5x5 kernel (stride 1) and 50 output channels, *FC1* has size 800x500, *FC2* has size 500x10, and the *MaxPool* layers have a 2x2 kernel (stride 2). This is roughly the LeNet5 architecture, but with one less fully connected layer at the end and around 10x the number of parameters. The SATNet layer contains 300 auxiliary variables, with $n = 729$ and $m = 600$. The full model is trained using Adam for 100 epochs using batch size 40, with a learning rate of $2 \times 10^{-3}$ for the SATNet layer and $1 \times 10^{-5}$ for the CNN.

## C.2  MNIST Mapping Problem

We use batch size 64 for training throughout all the experiments. We use the Sudoku CNN described above in Appendix Section C.1 as the backbone layer for all the experiments, except the one in Finding 4 where we vary the architecture. We use $m = 200, aux = 100$ for the SATNet layer for all the experiments, except the one in Finding 1 where we vary $m$ and $aux$.

Non-SATNet baseline: The whole network was trained with Adam using a $2 \times 10^{-3}$ learning rate.

Finding 1: The SATNet layer was trained with a $2 \times 10^{-3}$ learning rate, and the backbone layer was trained with a $1 \times 10^{-5}$ learning rate, both using Adam as was done above in Appendix Section C.1.

Finding 2: Both the SATNet layer and the backbone layer were trained with Adam.

Findings 3 and 4: The SATNet layer was trained with a $1 \times 10^{-3}$ learning rate using Adam, and the backbone layer was trained with a $1 \times 10^{-1}$ learning rate with SGD.

# D  More Experimental Results for the MNIST Mapping Problem

## D.1  Non-SATNet Baseline

The training accuracy for the non-SATNet baseline is 72.4±13.4% (3).

## D.2  Experiment 1

Table 4: Effects of $m$ and $aux$ on Training and Test Accuracy

| $m$ | $aux$ | Training Accuracy | Test Accuracy |
|---|---|---|---|
| 20 | 50 | 86.7±8.4% (1) | 86.8±8.4% (1) |
| 40 | 50 | 95.6±0.3% (0) | 95.5±0.3% (0) |
| 60 | 50 | 95.7±0.3% (0) | 95.6±0.4% (0) |
| 80 | 50 | 96.2±0.2% (0) | 96.0±0.3% (0) |
| 20 | 100 | 82.2±8.4% (1) | 82.4±8.4% (1) |
| 40 | 100 | 85.9±8.3% (1) | 85.9±8.3% (1) |
| 60 | 100 | 95.3±0.5% (0) | 95.3±0.5% (0) |
| 80 | 100 | 95.1±0.2% (0) | 94.9±0.2% (0) |
| 20 | 200 | 43.9±13.5% (6) | 44.0±13.4% (6) |
| 40 | 200 | 59.6±13.3% (4) | 59.7±13.3% (4) |
| 60 | 200 | 60.0±13.4% (4) | 60.2±13.3% (4) |
| 80 | 200 | 94.7±0.3% (0) | 94.6±0.3% (0) |
| 100 | 200 | 86.3±8.4% (1) | 86.2±8.4% (1) |
| 100 | 400 | 44.8±12.5% (4) | 45.0±12.6% (4) |
| 100 | 600 | 25.6±7.7% (7) | 26.2±7.9% (7) |
| 100 | 800 | 35.1±10.3% (6) | 35.8±10.4% (6) |
| 200 | 200 | 96.2±0.1% (0) | 95.8±0.2% (0) |
| 200 | 400 | 45.6±12.9% (4) | 45.3±12.9% (4) |
| 200 | 600 | 62.4±11.5% (2) | 62.4±11.7% (2) |
| 200 | 800 | 32.7±10.4% (5) | 33.2±10.5% (5) |
| 400 | 200 | 96.4±0.2% (0) | 96.0±0.2% (0) |
| 400 | 400 | 92.1±4.2% (0) | 91.8±4.0% (0) |
| 400 | 600 | 62.8±13.5% (3) | 62.7±13.4% (3) |
| 400 | 800 | 69.3±12.8% (3) | 69.4±12.7% (3) |

## D.3  Experiment 2

Table 5: Effects of Different Learning Rates on the SATNet and Backbone Layer on Training Accuracy

| SATNet Layer | Backbone Layer Learning Rate | | |
|---|---|---|---|
| Learning Rate | $1\times10^{-3}$ | $1\times10^{-4}$ | $1\times10^{-5}$ |
| $1\times10^{-3}$ | 19.6±8.5% (9) | 90.4±8.8% (1) | 96.7±0.2% (0) |
| $1\times10^{-4}$ | 17.0±4.1% (8) | 74.9±8.8% (0) | 96.5±0.2% (0) |
| $1\times10^{-5}$ | 14.4±3.4% (9) | 31.8±7.1% (5) | 71.9±5.4% (0) |

## D.4  Experiment 3

The training accuracy rose from 96.7±0.2% (0) to 99.1±0.1% (0).

## D.5 Experiment 4

Table 6: Effects of Different Neural Architectures on Training Accuracy

| Architectures | Parameters | Backbone Output Layer | |
|---|---|---|---|
| | | Softmax | Sigmoid |
| LeNet [40] | 68,626 | 63.2±14.2% (4) | 99.1±0.0% (0) |
| Sudoku CNN | 860,780 | 99.1±0.1% (0) | **99.5±0.0% (0)** |
| ResNet18 [43] | 11,723,722 | 67.6±6.2% (0) | 97.4±0.4% (0) |

## D.6 Further Investigation into m and aux

One of the reviewers proposed setting $m$ and $aux$ according to the relationship $m = out + aux$, where $out$ is the number of output variables. In the case of the MNIST mapping problem, we observed that while not necessarily optimal, it can be a good rule of thumb.

Another reviewer suggested that Experiment 1 be re-run with smaller values of $aux$. We show the results of re-running Experiment 1 with 10x smaller $aux$ in Figure 7. We can observe that in this regime where $m$ is significantly higher than $aux$, larger $m$ and smaller $aux$ show a more muted benefit.

Figure 7: Both graphs show test accuracy on the MNIST mapping problem with the shaded interval representing the standard error.