[Reviews · NeurIPS 2020]

Review 1

Summary and Contributions: EDIT after author response: + I thank the authors for their thoughtful response, and am happy to see the authors will address the issues around the framing/tone of the paper. I also like the suggestion to include a discussion of the different definitions of end-to-end; this discussion is likely to be of great value to the community, given the overloaded definition of end-to-end. - However, I do disagree with some of the claims made in the author response surrounding the discussion of end-to-end, which gives me pause as to whether the ultimate discussion will cover all the necessary nuances. To address these claims/elaborate further -- * SATNet-specific: In cases where the logical rules aren't known, it's not possible to use a discrete solving module, since the rules need to be learned. That is, using a classifier and a separate Sudoku solver wouldn't work for the SATNet experiments, as the premise of the SATNet experiment was that the rules of Sudoku weren't known. (Of course, the rules of Sudoku itself actually *are* known in reality, but presumably this is meant to be a stand-in example for cases where relevant rules *aren't* actually known in reality.) * In general: The reason to embed logical and other structured layers in neural networks is therefore inductive bias, in the same way convolutional layers provide inductive bias for image domains. This allows learning to, e.g., be more data-efficient. See e.g. OptNet (Amos and Kolter, ICML 2017), work on differentiable learning of submodular models (Krause et al, NeurIPS 2017), and Neural ODEs (Chen et al, NeurIPS 2018 best paper). In all cases, the challenging part is (a) making these layers differentiable and (b) integrating them into deep networks so that gradients can properly flow through them. This gives rise to the definition of end-to-end that the SATNet paper was presumably operating under. - I also had not initially noticed that the starting number of aux variables the authors use is a bit high for the simple symbolic grounding benchmark (pointed out by R2), and I would want to see an evaluation with a smaller number of aux variables, as this makes more sense given the particulars of the problem. Given that my major concern was the tone of the paper, and the authors have addressed that, I have raised my score from 6 --> 7. However, I hope that the authors will incorporate the above points about end-to-end learning and address the issues with the m/aux experiments in the final version of the paper. ------------------------------------------------------- This paper provides an assessment and critical review of SATNet, a differentiable MAXSAT solving layer for deep networks. In particular, the authors: * Critique the design of the visual Sudoku experiment in the SATNet paper, describing that it leaks label information and therefore does not demonstrate SATNet’s ability to solve the symbol grounding problem. * Analyze the training dynamics of SATNet on the visual Sudoku problem, finding that the relative speeds at which the image recognition CNN and the MAXSAT solver train completely determine whether SATNet is successful at solving visual Sudoku or not. * Assess the ability of SATNet and other models to solve a simple instance of the symbol grounding problem (via an MNIST mapping problem benchmark constructed by the authors) * Analyze the training dynamics of SATNet on the MNIST mapping problem, in order to provide recommendations for parameter configurations that enable SATNet to train successfully.

Strengths: This paper presents a strong evaluation of the results, claims, and training dynamics of a well-known piece of previous work (SATNet), providing both valuable critiques and recommendations for making the previous work more useful. In general, I believe that work that attempts to reproduce and/or evaluate previous contributions is an extremely important and valuable contribution to the literature. From my perspective, the analysis of how to tune SATNet parameters is extremely valuable. Given the difficulty of tuning these parameters, I find the present analysis — namely, regarding the relationship between m and aux, the relationship between the backbone and SATNet layers, and the best choices of network optimization algorithms and final layers to use — to be useful and insightful to practitioners hoping to use SATNet. The discussion of the symbol grounding problem is also interesting, and the modification of the visual Sudoku experimental design to address the symbol grounding problem is well-described. The proposed MNIST mapping problem benchmark also seems valuable in assessing future models’ ability to solve the symbol grounding problem. The authors are clear and transparent about their experimental design, and use good practices in this design (e.g. running over multiple random seeds).

Weaknesses: While the content of this paper is strong, a major and significant weakness of this paper is in ascribing intent to SATNet and the SATNet authors that may not be correct. As a result, this paper “oversells” its contribution via a strawman argument that SATNet claims to solve the symbol grounding problem. In particular, in the introduction, the authors present a quote from the original SATNet paper (draft line 59) to argue that SATNet claims to “learn[] to assign logical variables (symbols) to images of digits (perceptual phenomena) without explicit supervision of this mapping” (draft line 63). However, this is not what the SATNet paper claims. In particular, the original SATNet paper does not use the terminology of symbol grounding, nor does it claim to solve the symbol grounding problem. Instead, the authors of the original SATNet paper describe that the purpose of the Sudoku experiments is to “learn the rules of the game and learn how to solve Sudoku puzzles in an end-to-end manner” and that the goal of the visual Sudoku experiment in particular is to “show that [they] can embed [the] differentiable [SATNet] solver into larger architectures.” Therefore, while the design of the original visual Sudoku experiment indeed does not demonstrate the solving of the symbol grounding problem, this does not so much seem to be a _flaw_ as much as an experimental design created for a different purpose. As I believe the strawman argument diminishes the objectivity of the review, major changes to the title and framing addressing the above concerns would significantly strengthen the paper. In particular, the paper may require significant restructuring in order to remove the framing of _challenging_ SATNet regarding the symbol grounding problem (as opposed to _assessing_ its ability to solve this problem).

Correctness: The technical claims and empirical demonstrations seem correct. (However, I do not believe the argument that SATNet claims to solve the symbol grounding problem is true.)

Clarity: The paper is very clear and easy to follow.

Relation to Prior Work: Yes, the authors adequately discuss the relationship to previous work.

Reproducibility: Yes

Additional Feedback: Regarding my overall rating: My rating is borderline largely because of the framing of the paper. However, I do believe the content in this paper is sound and of great value to the community. My rating would significantly increase if the framing of the paper were changed according to the feedback above. Regarding the broader impact statement: It could be good for the authors to additionally address why the symbol grounding problem is important. Minor typo (which does not affect my score): I believe there is a word missing somewhere in the sentence between lines 71--75.


Review 2

Summary and Contributions: The authors of the draft challenged SATNet, a differentiable MAXSAT solver, with its ability on symbolic grounding. The SATNet paper claimed that they could solve the sudoku problem with MNIST image inputs and one-hot encoding labels. However, the authors showed that, by masking out all the labels for the inputting cells and leaving only the outputting cells' supervision, the SATNet failed with a 0% training accuracy. Further, on some random seeds, SATNet failed when the inputting ConvNet failed. Thus, the authors concluded that SATNet couldn't perform symbolic grounding. To provide a benchmark to test the ability for symbolic grounding, the authors designed the MNIST mapping problem. The problem has 10 separated input and output neurons, with an MNIST classifier connected to the inputting neurons. The goal of the problem is to match the 10 input neurons to the 10 output neurons with an identity connection. The authors tested various configurations of SATNet on this model problem and provided suggestions for tuning hyper-parameters of the SATNet problem.

Strengths: The authors provided a through investigation on SATNet's ability to perform symbol grounding. Further, they provide a model problem to check the symbol grounding ability, and also guides on tuning the hyper-parameter of the SATNet layer in their model problem.

Weaknesses: The embarrassing problem is that, as the reviewer understands, the SATNet paper didn't claim to solve the symbol grounding problem, and it wasn't meant to perform symbolic grounding. As the name suggests, it is simply a differentiable (MAX)SAT solver capable of learning the clause weights. So in the image sudoku problem, it was intended to allow the label supervision to pass through for the underlying ConvNet layer. It is not a leakage since the prediction doesn't use the label information, and SATNet wasn't meant to constraint itself to perform symbolic grounding. Before SATNet, it is not even possible to accept probabilistic inputs (from NN) to an SAT problem and let the supervision flow back to the inputting neurons. That is the reason that the SATNet paper claimed to "integrate logical structures within DL." On the random seed issue, the MNIST sudoku problem is indeed very picky on the quality of the inputting ConvNet. It is because the paper only counts a prediction as correct if all bits are entirely correct. In the draft's experiment, the bad random seed resulted in a ~90% testing accuracy for the inputting ConvNet, so the probability that every inputting bits on the board are correct is ~2% (from 0.9^36.2). Given that the inputs are already bad, the output from the SATNet layer will only be worse. Thus, the random seed issue is more of a problem with the evaluation metric. If we look at bit-wise accuracy, it would be much more tolerant. But this is indeed an issue for the SAT problem with noisy input. On the benchmark problem (MNIST mapping), since the optimal connection from the input to the output is the identity, it wouldn't be surprising that a simple dense layer performs well. Because we only have 10 input and output variables, it is natural that increasing the number of aux variable to >200 will harm the performance since it controls the complexity of the SAT problem, while the underlying ground truth is very simple. I guess the curves will be pretty robust when the number of aux variables is not that large. Also, the parameter m controls the clause matrix's rank instead of the "number of clauses." Setting it to the total number of variables is usually safe for small problems.

Correctness: The evaluations are solid on the purpose of examining the symbol grounding ability, but the main claim is wrong since the SATNet paper didn't claim to solve the symbol grounding problem.

Clarity: The paper is well-written.

Relation to Prior Work: The authors did a proper introduction to the prior works.

Reproducibility: Yes

Additional Feedback: ==== After reading the authors' feedback ==== I agree that the confusion happens because there's a different notion of "end-to-end learning" that we are not aware of before. A careful discussion will help if the authors change the tone/framing, as they said in the response. BTW, my suggestion on setting m to the number of variables means the total number of variables, that is, m=out+aux. If the number of clause m is much smaller than that, there are fewer clauses than the variables. An SAT problem with fewer clauses than variables means some variables are either free or not used (and can be removed), so it either has no solution or exp(num_var - m) solutions. Most non-trivial SAT/MAXSAT instances have at least one clauses for each variable so that they are well-defined. Specifically for SATNet, having m >= out + aux means having a full-rank clauses matrix to have the full expressive power. The author's experiment somewhat also verifies these arguments. We can verify in Figure 4 that a proper m always leads to almost perfect accuracies. Also, I would like to note that the visual sudoku without label information will be *much* harder than a sudoku problem and maybe even ill-defined. If we mask the label information, there's *no* direct symbol grounding or supervision. Whenever there's an MNIST image there will be no label, and the labels are only given to the blank image. What is worse, we do not have any locality information in the settings, and inputs are considered bit strings without positional hints. I really cannot think of how we can train it, using any architecture, in 9k examples. Given the above reasons, I decided to bump my score from 4 to 5.


Review 3

Summary and Contributions: The paper gives a critical review of SATNet and provides practical solutions for applying SATNet more effectively. First, authors find that the labels are leaked for pre-filled cells in the experiment for visual Sudoku, and SATNet fails for visual Sudoku after the flaw had been rectified. Second, authors find SATNet is sensitive to specific combinations of hyperparameters, optimizers, and neural architectures on the proposed simple test, MNIST Mapping Problem.

Strengths: + The paper presents a critical review of SATNet that has won conference best paper honorable mention. I appreciate for authors’ critical thinking and detailed descriptions. + Authors find SATNet completely fails for visual Sudoku, which is pinpointed to SATNet’s inability for symbol grounding problem. + Authors present a test, MNIST Mapping Problem, for evaluating SATNet on the combination of two simple problems for symbol grounding and logical reasoning, respectively. Besides, they provide some insights for applying SATNet more effectively. + The authors found that the label leakage leads to a two-step training process, and Figure 3 provide the evidence.

Weaknesses: + The paper is lack of novelty. It only presents a critical review and does not further discuss the theoretical solutions or insights for the problems. + The contribution of the paper is somewhat limited. It only criticizes one part of a paper, i.e., the ability of SATNet for joint symbol grounding and logical reasoning. + Are the practical solutions for applying SATNet effectively for other deep neural networks and applications? This needs to be discussed further. Besides, are there any insights or explainable reasons for the settings of hyperparameters, optimizer, and neural architectures? ----------------------- The authors have well answered my main concern.

Correctness: Yes

Clarity: well written

Relation to Prior Work: Yes

Reproducibility: Yes

Additional Feedback:


Review 4

Summary and Contributions: I have read the author response and reviewer discussion. The paper identifies a bug in the implementation of SATNet that partially invalidates the main claim of the SATNet paper. The main claim of SATNet is that it can integrate symbolic and sub-symbolic reasoning "Integrating logical reasoning within deep learning architectures has been a major goal of modern AI systems. In this paper, we propose a new direction toward this goal by introducing a differentiable (smoothed) maximum satisfiability (MAXSAT) solver that can be integrated into the loop of larger deep learning systems." Specifically the authors show that the MNIST digit labels are leaked for the observed digits in the SATNet visual Sudoku experiment, and that without these leaked labels, SATNet fails to solve the visual Sudoku puzzle. The paper further describes four "tricks" that help the training of SATNet and introduce a simple benchmark for testing symbolic grounding. The authors then show that by applying these tricks they can train SATNet to consistently solve the simple benchmark. However, they still fail to train SATNet to solve visual Sudoku, a more challenging task. I write that the paper only partially invalidates the main claim of the SATNet paper since the authors of this paper show that SATNet can indeed solve the simple symbolic grounding benchmark they propose, using the tricks they describe. As such they show that SATNet can "Integrate logical reasoning within deep learning architectures" however, only on simple problems, and not the visual Sudoku problem as claimed. The SATNet paper was published in a top-tier conference (ICLR) and won a best paper honorable mention award, so it's a fairly "prominent" paper. As such I believe the negative findings of this paper is important to be published in a similarly prominent venue. The paper is well written and the empirical evaluations seem sound. As such I recommend this paper for acceptance.

Strengths: Important finding and well written. Thorough empirical evaluations.

Weaknesses: The paper only partially invalidates the main claim of the SATNet paper, which diminishes the significance.

Correctness: To the best of my knowledge.

Clarity: Yes, the paper is well written. The main point is well argued.

Relation to Prior Work: The authors note that efforts to combine symbolic and sub-symbolic approaches fall in two categories: "soft logic operators" and "inductive logic programming". However, this completely ignores the entirely sub-symbolic reasoning approaches, e.g. "A simple neural network module for relational reasoning", "Recurrent Relational Networks", and "Learning a SAT Solver from Single-Bit Supervision". I would encourage the authors to include this line of works in their revised paper.

Reproducibility: Yes

Additional Feedback: I would reduce the amount of adjectives used. It reads a little "sensational", but this might be personal preference.

[Author Response · NeurIPS 2020]

We thank the reviewers for their insightful comments on our work, and appreciate their help in making our manuscript
better. Below, we address the major concerns that they have brought up.

**Tone/Framing of the Paper:** We had argued that there was a flaw in the SATNet authors' visual Sudoku experimental
design, primarily because they claimed to have integrated logical reasoning into a deep neural network to enable
"end-to-end" learning in a "minimally supervised fashion." We understand "end-to-end" learning to mean that there
are no intermediate labels that need to be collected. For example, end-to-end speech recognition means that there is
no need to collect intermediate phoneme labels to build an acoustic model, and end-to-end facial recognition means
there is no need to collect intermediate labels for a nose/eyes/lips detector. We do not argue that SATNet has claimed to
solve the symbol grounding problem in general, only that a necessary implication of their claim to end-to-end learning
for visual Sudoku is that they must have solved it for this specific instance. We are concerned that many papers about
logical reasoning in deep learning omit any discussion about symbol grounding, and we hope that our paper reminds
the community of its importance. Ultimately, if neural networks are not able to ground perceptual phenomena into
symbols, there is little need to integrate any logical reasoning into deep learning, since we can always build a system
that has distinct machine learning and discrete solving modules, like the OpenAI Rubik's Cube solver.

That said, we appreciate Reviewers 1 and 2 for suggesting that there is an alternative interpretation of "end-to-end"
learning that means being able to train a deep network with gradient descent by making every component in the
network differentiable, which might well have been the SATNet authors' intended claim. This is something that
we had not previously considered, which likely made our argument overly critical towards SATNet. To allay the
reviewers' concerns, we will amend our title to Reviewer 1's proposal: "Assessing SATNet's Ability to Solve the
Symbol Grounding Problem". We will further remove any reference to SATNet's use of intermediate labels as a "flaw"
or a "bug", and provide a discussion about the two different interpretations of end-to-end learning. It is quite possible
that many readers of the original SATNet paper might have understood it with the same interpretation as Reviewer 4
and us, and thus, we think that our work will clarify to the community the exact nature of SATNet's capabilities. We
empathize with the reviewers' concerns that our work might come across as "sensational" or lacking "objectivity", and
welcome further suggestions that can improve the quality of our manuscript.

**Significance of our Contribution:** Reviewer 3 was concerned about a lack of novelty in our work besides our critical
review of SATNet's visual Sudoku experiment. One of the main contributions of our work is to highlight the difference
between defining gradients for an architecture (the end-to-end learning interpretation of Reviewers 1 and 2) and actually
being able to train it. This problem is similar to the one that the deep learning community had to solve for neural
networks to enable the training of architectures with hundreds of layers, where gradients are well-defined but successful
training is non-trivial and requires careful initialization, batch normalization, adaptive learning rates, etc. Our work
identifies several factors that affect the learning dynamics of SATNet and provides practical suggestions for configuring
SATNet to enable successful training. We reveal surprising complexities that are unique to SATNet and break standard
deep learning norms. For example, using different learning rates for different layers in neural networks is not a common
practice, since the use of Adam usually suffices. But for the case of SATNet, even when Adam is used, the backbone
layer has to learn at a slower rate than the SATNet layer for successful training to occur. In addition, our findings go
beyond verifying simple intuitions about SATNet. We believe that broad statements like setting $m$ to "the total number
of variables," as Reviewer 2 suggests, do not suffice to elucidate the complicated relationship between $m$ and $aux$, and
their effect on the intricate training dynamics in SATNet. For example, we see in Figure 4 that even when $m = 20$ is
the total number of variables, performance can degrade for certain values of $aux$, but not when $m$ is increased. Instead
of relying on intuition, our work systematically studies the empirical effects of different SATNet configurations and
highlights key steps required for successful SATNet training to occur.

**Other Comments:** We thank Reviewer 4 for pointing out relevant prior work from sub-symbolic reasoning, and will
include them in our manuscript. Reviewer 2 suggests that SATNet's failure on some random seeds was due to the
evaluation metric for visual Sudoku, but we showed that the failure also occurred on the much simpler MNIST mapping
problem, where the metric is the percentage of test images classified accurately. It is clear that these cases should be
identified as learning failures (resulting from the sensitivity of SATNet to the random initialization) and not dismissed
as mere artifacts of the evaluation metric. As Reviewer 2 points out, $m$ is strictly speaking the number of clauses that
the SATNet layer *represents*; we stick to the same terminology used in the original SATNet paper and believe that most
readers will not be confused by this.

**Concluding Remarks:** In general, we think that the differences between deep learning and logic mirror the ones
between continuous and discrete optimization. These differences go far deeper than the superficial lack of derivatives in
discrete optimization, and we believe true progress has to come from significantly tighter integrations between deep
learning and logic. We are excited that our work brings these differences to the forefront and encourages the community
to think more critically about how to go about integrating logical reasoning into deep learning.

[Meta-Review · NeurIPS 2020]

This paper is a critical follow-up to the SATNet paper at ICLR last year which introduced a differentiable MAXSAT solver (and won a best paper award). The paper is somewhat critcal of some of the claims of the SATNet paper and proposes some "tricks" for overcoming these problems. Strengths identified by the reviewers include that the improvements to SATNet will be practically useful and that the analysis and critiques are insightful. Weaknesses are that some of the claimed problems with SATNet seem to be more a misunderstanding of or ambiguity in what was claimed in that paper, as opposed to fundamental issues. The authors addressed these issues in the rebuttal and promise to tone down the claims. The AC also read the paper and agrees that the paper should be accepted but authors must make the claims and framing clearer in the camera ready.